# Q-LEARNING WITH UCB EXPLORATION IS SAMPLE EFFICIENT FOR INFINITE-HORIZON MDP

**Kefan Dong**[*]**, Yuanhao Wang**[*]
Institute for Interdisciplinary Information Sciences,
Tsinghua University
`{dkf16,yuanhao-16}@mails.tsinghua.edu.cn`

**Xiaoyu Chen**
Key Laboratory of Machine Perception, MOE, School of EECS,
Peking University
`cxy30@pku.edu.cn`

**Liwei Wang**
Key Laboratory of Machine Perception, MOE, School of EECS
Center for Data Science, Peking University
`wanglw@cis.pku.edu.cn`

## ABSTRACT

A fundamental question in reinforcement learning is whether model-free algorithms are sample efficient. Recently, Jin et al. (2018) proposed a Q-learning algorithm with UCB exploration policy, and proved it has nearly optimal regret bound for finite-horizon episodic MDP. In this paper, we adapt Q-learning with UCB-exploration bonus to infinite-horizon MDP with discounted rewards *without* accessing a generative model. We show that the *sample complexity of exploration* of our algorithm is bounded by $\tilde{O}(\frac{SA}{\epsilon^2(1-\gamma)^7})$. This improves the previously best known result of $\tilde{O}(\frac{SA}{\epsilon^4(1-\gamma)^8})$ in this setting achieved by delayed Q-learning (Strehl et al., 2006), and matches the lower bound in terms of $\epsilon$ as well as $S$ and $A$ up to logarithmic factors.

## 1 INTRODUCTION

The goal of reinforcement learning (RL) is to construct efficient algorithms that learn and plan in sequential decision making tasks when the underlying system dynamics are unknown. A typical model in RL is Markov Decision Process (MDP). At each time step, the environment is in a state $s$. The agent takes an action $a$, obtain a reward $r$, and then the environment transits to another state. In reinforcement learning, the transition probability distribution is unknown. The algorithm needs to learn the transition dynamics of MDP, while aiming to maximize the cumulative reward. This poses the exploration-exploitation dilemma: whether to act to gain new information (explore) or to act consistently with past experience to maximize reward (exploit).

Theoretical analyses of reinforcement learning fall into two broad categories: those assuming a simulator (a.k.a. generative model), and those without a simulator. In the first category, the algorithm is allowed to query the outcome of any state action pair from an oracle. The emphasis is on the number of calls needed to estimate the $Q$ value or to output a near-optimal policy. There has been extensive research in literature following this line of research, the majority of which focuses on discounted infinite horizon MDPs (Azar et al., 2011; Even-Dar & Mansour, 2003; Sidford et al., 2018b). The current results have achieved near-optimal time and sample complexities (Sidford et al., 2018b;a).

---

[*]These two authors contributed equally.

Without a simulator, there is a dichotomy between finite-horizon and infinite-horizon settings. In finite-horizon settings, there are straightforward definitions for both regret and sample complexity; the latter is defined as the number of samples needed *before* the policy becomes near optimal. In this setting, extensive research in the past decade (Jin et al., 2018; Azar et al., 2017; Jaksch et al., 2010; Dann et al., 2017) has achieved great progress, and established nearly-tight bounds for both regret and sample complexity.

The infinite-horizon setting is a very different matter. First of all, the performance measure cannot be a straightforward extension of the sample complexity defined above (See Strehl & Littman (2008) for detailed discussion). Instead, the measure of sample efficiency we adopt is the so-called *sample complexity of exploration* (Kakade et al., 2003), which is also a widely-accepted definition. This measure counts the number of times that the algorithm "makes mistakes" along the whole trajectory. See also (Strehl & Littman, 2008) for further discussions regarding this issue.

Several model based algorithms have been proposed for infinite horizon MDP, for example R-max (Brafman & Tennenholtz, 2003), MoRmax (Szita & Szepesvári, 2010) and UCRL-$\gamma$ (Lattimore & Hutter, 2012). It is noteworthy that there still exists a considerable gap between the state-of-the-art algorithm and the theoretical lower bound (Lattimore & Hutter, 2012) regarding $1/(1 - \gamma)$ factor.

Though model-based algorithms have been proved to be sample efficient in various MDP settings, most state-of-the-art RL algorithms are developed in the model-free paradigm (Schulman et al., 2015; Mnih et al., 2013; 2016). Model-free algorithms are more flexible and require less space, which have achieved remarkable performance on benchmarks such as Atari games and simulated robot control problems.

For infinite horizon MDPs without access to simulator, the best model-free algorithm has a sample complexity of exploration $\tilde{\mathcal{O}}(\frac{SA}{\epsilon^4(1-\gamma)^8})$, achieved by delayed Q-learning (Strehl et al., 2006). The authors provide a novel strategy of argument when proving the upper bound for the sample complexity of exploration, namely identifying a sufficient condition for optimality, and then bound the number of times that this condition is violated.

However, the results of Delayed Q-learning still leave a quadratic gap in $1/\epsilon$ from the best-known lower bound. This is partly because the updates in Q-value are made in an over-conservative way. In fact, the loose sample complexity bound is a result of delayed Q-learning algorithm itself, as well as the mathematical artifact in their analysis. To illustrate this, we construct a hard instance showing that Delayed Q-learning incurs $\Omega(1/\epsilon^3)$ sample complexity. This observation, as well as the success of the Q-learning with UCB algorithm (Jin et al., 2018) in proving a regret bound in finite-horizon settings, motivates us to incorporate a UCB-like exploration term into our algorithm.

In this work, we propose a Q-learning algorithm with UCB exploration policy. We show the sample complexity of exploration bound of our algorithm is $\tilde{\mathcal{O}}(\frac{SA}{\epsilon^2(1-\gamma)^7})$. This strictly improves the previous best known result due to Delayed Q-learning. It also matches the lower bound in the dependence on $\epsilon$, $S$ and $A$ up to logarithmic factors.

We point out here that the infinite-horizon setting *cannot* be solved by reducing to finite-horizon setting. There are key technical differences between these two settings: the definition of sample complexity of exploration, time-invariant policies and the error propagation structure in Q-learning. In particular, the analysis techniques developed in (Jin et al., 2018) do not directly apply here. We refer the readers to Section 3.2 for detailed explanations and a concrete example.

The rest of the paper is organized as follows. After introducing the notation used in the paper in Section 2, we describe our infinite Q-learning with UCB algorithm in Section 3. We then state our main theoretical results, which are in the form of PAC sample complexity bounds. In Section 4 we present some interesting properties beyond sample complexity bound. Finally, we conclude the paper in Section 5.

## 2 PRELIMINARY

We consider a Markov Decision Process defined by a five tuple $\langle \mathcal{S}, \mathcal{A}, p, r, \gamma \rangle$, where $\mathcal{S}$ is the state space, $\mathcal{A}$ is the action space, $p(s'|s, a)$ is the transition function, $r : \mathcal{S} \times \mathcal{A} \to [0, 1]$ is the deterministic

reward function, and $0 \leq \gamma < 1$ is the discount factor for rewards. Let $S = |\mathcal{S}|$ and $A = |\mathcal{A}|$ denote the number of states and the number of actions respectively.

Starting from a state $s_1$, the agent interacts with the environment for infinite number of time steps. At each time step, the agent observes state $s_t \in \mathcal{S}$, picks action $a_t \in \mathcal{A}$, and receives reward $r_t$; the system then transits to next state $s_{t+1}$.

Using the notations in Strehl et al. (2006), a policy $\pi_t$ refers to the non-stationary control policy of the algorithm since step $t$. We use $V^{\pi_t}(s)$ to denote the value function under policy $\pi_t$, which is defined as $V^{\pi_t}(s) = \mathbb{E}[\sum_{i=1}^{\infty} \gamma^{i-1} r(s_i, \pi_{t+i-1}(s_i))|s_1 = s]$. We also use $V^*(s) = \sup_{\pi} V^{\pi}(s)$ to denote the value function of the optimal policy. Accordingly, we define $Q^{\pi_t}(s,a) = r(s,a) + \mathbb{E}[\sum_{i=2}^{\infty} \gamma^{i-1} r(s_i, \pi_{t+i-1}(s_i))|s_1 = s, a_1 = a]$ as the Q function under policy $\pi_t$; $Q^*(s,a)$ is the Q function under optimal policy $\pi^*$.

We use the sample complexity of exploration defined in Kakade et al. (2003) to measure the learning efficiency of our algorithm. This sample complexity definition has been widely used in previous works Strehl et al. (2006); Lattimore & Hutter (2012); Strehl & Littman (2008).

**Definition 1. Sample complexity of Exploration** *of an algorithm $\mathcal{ALG}$ is defined as the number of time steps $t$ such that the non-stationary policy $\pi_t$ at time $t$ is not $\epsilon$-optimal for current state $s_t$, i.e. $V^{\pi_t}(s_t) < V^*(s_t) - \epsilon$.*

Roughly speaking, this measure counts the number of mistakes along the whole trajectory. We use the following definition of PAC-MDP Strehl et al. (2006).

**Definition 2.** *An algorithm $\mathcal{ALG}$ is said to be* **PAC-MDP** *(Probably Approximately Correct in Markov Decision Processes) if, for any $\epsilon$ and $\delta$, the sample complexity of $\mathcal{ALG}$ is less than some polynomial in the relevant quantities $(S, A, 1/\epsilon, 1/\delta, 1/(1-\gamma))$, with probability at least $1 - \delta$.*

Finally, recall that Bellman equation is defined as the following:

$$\begin{cases} V^{\pi_t}(s) = Q^{\pi_t}(s, \pi_t(s)) \\ Q^{\pi_t}(s,a) := (r_t + \gamma \mathbb{P} V^{\pi_{t+1}})(s,a), \end{cases} \qquad \begin{cases} V^*(s) = Q^*(s, \pi^*(s)) \\ Q^*(s,a) := (r_t + \gamma \mathbb{P} V^*)(s,a), \end{cases}$$

which is frequently used in our analysis. Here we denote $[\mathbb{P} V^{\pi_t}](s,a) := \mathbb{E}_{s' \sim p(\cdot|s,a)} V^{\pi_{t+1}}(s')$.

## 3 MAIN RESULTS

In this section, we present the UCB Q-learning algorithm and the sample complexity bound.

### 3.1 ALGORITHM

---

**Algorithm 1** Infinite Q-learning with UCB

---

Parameters: $\epsilon, \gamma, \delta$

Initialize $Q(s,a), \hat{Q}(s,a) \leftarrow \frac{1}{1-\gamma}, N(s,a) \leftarrow 0, \epsilon_1 \leftarrow \frac{\epsilon}{24RM \ln \frac{1}{1-\gamma}}, H \leftarrow \frac{\ln 1/((1-\gamma)\epsilon_1)}{\ln 1/\gamma}$.

Define $\iota(k) = \ln(SA(k+1)(k+2)/\delta), \alpha_k = \frac{H+1}{H+k}$.

**for** $t = 1, 2, \ldots$ **do**

5:    Take action $a_t \leftarrow \arg\max_{a'} \hat{Q}(s_t, a')$

    Receive reward $r_t$ and transit to $s_{t+1}$

    $N(s_t, a_t) \leftarrow N(s_t, a_t) + 1$

    $k \leftarrow N(s_t, a_t), b_k \leftarrow \frac{c_2}{1-\gamma}\sqrt{\frac{H\iota(k)}{k}}$             $\triangleright$ $c_2$ is a constant and can be set to $4\sqrt{2}$

    $\hat{V}(s_{t+1}) \leftarrow \max_{a \in A} \hat{Q}(s_{t+1}, a)$

10:   $Q(s_t, a_t) \leftarrow (1-\alpha_k)Q(s_t, a_t) + \alpha_k \left[ r(s_t, a_t) + b_k + \gamma \hat{V}(s_{t+1}) \right]$

    $\hat{Q}(s_t, a_t) \leftarrow \min(\hat{Q}(s_t, a_t), Q(s_t, a_t))$

**end for**

---

Here $c_2 = 4\sqrt{2}$ is a constant. $R = \lceil \ln \frac{3}{\epsilon(1-\gamma)}/(1-\gamma) \rceil$, while the choice of $M$ can be found in Section. 3.3. $(M = \mathcal{O}(\ln 1/((1-\gamma)\epsilon)))$. The learning rate is defined as $\alpha_k = (H+1)/(H+k)$. $H$ is chosen as $\frac{\ln 1/((1-\gamma)\epsilon_1)}{\ln 1/\gamma}$, which satisfies $H \leq \frac{\ln 1/((1-\gamma)\epsilon_1)}{1-\gamma}$.

Our UCB Q-learning algorithm (Algorithm 1) maintains an optimistic estimation of action value function $Q(s, a)$ and its historical minimum value $\hat{Q}(s, a)$. $N_t(s, a)$ denotes the number of times that $(s, a)$ is experienced before time step $t$; $\tau(s, a, k)$ denotes the time step $t$ at which $(s_t, a_t) = (s, a)$ for the $k$-th time; if this state-action pair is not visited that many times, $\tau(s, a, k) = \infty$. $Q_t(s, a)$ and $\hat{Q}_t(s, a)$ denotes the $Q$ and $\hat{Q}$ value of $(s, a)$ that the algorithm maintains when arriving at $s_t$ respectively.

## 3.2 SAMPLE COMPLEXITY OF EXPLORATION

Our main result is the following sample complexity of exploration bound.

**Theorem 1.** *For any $\epsilon > 0$, $\delta > 0, 1/2 < \gamma < 1$, with probability $1 - \delta$, the sample complexity of exploration (i.e., the number of time steps $t$ such that $\pi_t$ is not $\epsilon$-optimal at $s_t$) of Algorithm 1 is at most*

$$\tilde{\mathcal{O}}\left(\frac{SA \ln 1/\delta}{\epsilon^2 (1 - \gamma)^7}\right),$$

*where $\tilde{\mathcal{O}}$ suppresses logarithmic factors of $1/\epsilon$, $1/(1 - \gamma)$ and $SA$.*

We first point out the obstacles for proving the theorem and reasons why the techniques in Jin et al. (2018) do not directly apply here. We then give a high level description of the ideas of our approach.

One important issue is caused by the difference in the definition of sample complexity for finite and infinite horizon MDP. In finite horizon settings, sample complexity (and regret) is determined in the first $T$ timesteps, and only measures the performance at the initial state $s_1$ (i.e. $(V^* - V^\pi)(s_1)$). However, in the infinite horizon setting, the agent may enter under-explored regions at any time period, and sample complexity of exploration characterizes the performance at all states the agent enters.

The following example clearly illustrates the key difference between infinite-horizon and finite-horizon. Consider an MDP with a starting state $s_1$ where the probability of leaving $s_1$ is $o(T^{-1})$. In this case, with high probability, it would take more than $T$ timesteps to leave $s_1$. Hence, guarantees about the learning in the first $T$ timesteps or about the performance at $s_1$ imply almost nothing about the number of mistakes the algorithm would make in the rest of the MDP (i.e. the *sample complexity of exploration* of the algorithm). As a result, the analysis for finite horizon MDPs cannot be directly applied to infinite horizon setting.

This calls for techniques for counting mistakes along the entire trajectory, such as those employed by Strehl et al. (2006). In particular, we need to establish convenient sufficient conditions for being $\epsilon$-optimal at timestep $t$ and state $s_t$, i.e. $V^*(s_t) - V^{\pi_t}(s_t) \leq \epsilon$. Then, bounding the number of violations of such conditions gives a bound on sample complexity.

Another technical reason why the proof in Jin et al. (2018) cannot be directly applied to our problem is the following: In finite horizon settings, Jin et al. (2018) decomposed the learning error at episode $k$ and time $h$ as errors from a set of *consecutive* episodes before $k$ at time $h + 1$ using a clever design of learning rate. However, in the infinite horizon setting, this property does not hold. Suppose at time $t$ the agent is at state $s_t$ and takes action $a_t$. Then the learning error at $t$ only depends on those previous time steps such that the agent encountered the same state as $s_t$ and took the same action as $a_t$. Thus the learning error at time $t$ cannot be decomposed as errors from a set of *consecutive* time steps before $t$, but errors from a set of *non-consecutive* time steps without any structure. Therefore, we have to control the sum of learning errors over an unstructured set of time steps. This makes the analysis more challenging.

Now we give a brief road map of the proof of Theorem 1. Our first goal is to establish a sufficient condition so that $\pi_t$ learned at step $t$ is $\epsilon$-optimal for state $s_t$. As an intermediate step we show that a sufficient condition for $V^*(s_t) - V^{\pi_t}(s_t) \leq \epsilon$ is that $V^*(s_{t'}) - Q^*(s_{t'}, a_{t'})$ is small for a few time steps $t'$ within an interval $[t, t + R]$ for a carefully chosen $R$ (Condition 1). Then we show the desired sufficient condition (Condition 2) implies Condition 1. We then bound the total number of bad time steps on which $V^*(s_t) - Q^*(s_t, a_t)$ is large for the whole MDP; this implies a bound on the number of violations of Condition 2. This in turn relies on a key technical lemma (Lemma 2).

The remaining part of this section is organized as follows. We establish the sufficient condition for $\epsilon$-optimality in Section 3.3. The key lemma is presented in Section 3.4. Finally we prove Theorem 1 in Section 3.5.

### 3.3 SUFFICIENT CONDITION FOR $\epsilon$-OPTIMALITY

In this section, we establish a sufficient condition (Condition 2) for $\epsilon$-optimality at time step $t$.

For a fixed $s_t$, let TRAJ($R$) be the set of length-$R$ trajectories starting from $s_t$. Our goal is to give a sufficient condition so that $\pi_t$, the policy learned at step $t$, is $\epsilon$-optimal. For any $\epsilon_2 > 0$, define $R := \lceil \ln \frac{1}{\epsilon_2(1-\gamma)}/(1-\gamma) \rceil$. Denote $V^*(s_t) - Q^*(s_t, a_t)$ by $\Delta_t$. We have

$$
\begin{aligned}
&V^*(s_t) - V^{\pi_t}(s_t) \\
=&V^*(s_t) - Q^*(s_t, a_t) + Q^*(s_t, a_t) - V^{\pi_t}(s_t) \\
=&V^*(s_t) - Q^*(s_t, a_t) + \gamma \mathbb{P}\left(V^* - V^{\pi_t}\right)(s_t, \pi_t(s_t)) \\
=&V^*(s_t) - Q^*(s_t, a_t) + \gamma \sum_{s_{t+1}} p\left(s_{t+1}|s_t, \pi_t(s_t)\right) \cdot \left[V^*(s_{t+1}) - Q^*(s_{t+1}, a_{t+1})\right] + \\
&\gamma \sum_{s_{t+1}, s_{t+2}} p\left(s_{t+2}|s_{t+1}, \pi_{t+1}(s_{t+1})\right) \cdot p\left(s_{t+1}|s_t, \pi_t(s_t)\right) \left[V^*(s_{t+2}) - Q^*(s_{t+2}, a_{t+2})\right] \\
&\cdots \\
\leq & \epsilon_2 + \sum_{\substack{traj\in \\ \text{TRAJ}(R)}} p(traj) \cdot \left[\sum_{j=0}^{R-1} \gamma^j \Delta_{t+j}\right],
\end{aligned} \tag{1}
$$

where the last inequality holds because $\frac{\gamma^R}{1-\gamma} \leq \epsilon_2$, which follows from the definition of $R$.

For any fixed trajectory of length $R$ starting from $s_t$, consider the sequence $(\Delta_{t'})_{t \leq t' < t+R}$. Let $X_t^{(i)}$ be the $i$-th largest item of $(\Delta_{t'})_{t \leq t' < t+R}$. Rearranging Eq. (1), we obtain

$$
V^*(s_t) - V^{\pi_t}(s_t) \leq \epsilon_2 + E_{traj}\left[\sum_{i=1}^R \gamma^{i-1} X_t^{(i)}\right]. \tag{2}
$$

We first prove that Condition 1 implies $\epsilon$-optimality at time step $t$ when $\epsilon_2 = \epsilon/3$.

**Condition 1.** *Let $\xi_i := \frac{1}{2^{i+2}} \epsilon_2 \left(\ln \frac{1}{1-\gamma}\right)^{-1}$. For all $0 \leq i \leq \lfloor \log_2 R \rfloor$,*

$$
E[X_t^{(2^i)}] \leq \xi_i. \tag{3}
$$

**Claim 1.** *If Condition 1 is satisfied at time step $t$, the policy $\pi_t$ is $\epsilon$-optimal at state $s_t$, i.e. $V^*(s_t) - V^{\pi_t}(s_t) \leq \epsilon$.*

*Proof.* Note that $X_t^{(i)}$ is monotonically decreasing with respect to $i$. Therefore, $E[X_t^{(i)}] \leq E[X_t^{(2^{\lfloor \log_2 i \rfloor})}]$. Eq. (3) implies that for $1/2 < \gamma < 1$,

$$
\begin{aligned}
E\left[\sum_{i=1}^R \gamma^{i-1} X_t^{(i)}\right] &= \sum_{i=1}^R \gamma^{i-1} E[X_t^{(i)}] \leq \sum_{i=1}^R \gamma^{i-1} E[X_t^{(2^{\lfloor \log_2 i \rfloor})}] \\
&\leq \sum_{i=1}^R \gamma^{i-1} 2^{-\lfloor \log_2 i \rfloor - 2} \epsilon_2 \left(\ln \frac{1}{1-\gamma}\right)^{-1} \leq \sum_{i=1}^R \frac{\gamma^{i-1}}{i} \epsilon_2 \left(\ln \frac{1}{1-\gamma}\right)^{-1} \leq 2\epsilon_2,
\end{aligned}
$$

where the last inequality follows from the fact that $\sum_{i=1}^\infty \frac{\gamma^{i-1}}{i} = \frac{1}{\gamma} \ln \frac{1}{1-\gamma}$ and $\gamma > 1/2$.

Combining with Eq. 2, we have, $V^*(s_t) - V^{\pi_t}(s_t) \leq \epsilon_2 + E\left[\sum_{i=1}^R \gamma^{i-1} X_t^{(i)}\right] \leq 3\epsilon_2 = \epsilon$. $\qquad \square$

Next we show that given $i, t$, Condition 2 implies Eq. (3).

**Condition 2.** *Define* $L = \lfloor \log_2 R \rfloor$. *Let* $M = \max\left\{\lceil 2\log_2 \frac{1}{\xi_L(1-\gamma)} \rceil, 10\right\}$, *and* $\eta_j = \frac{\xi_i}{M} \cdot 2^{j-1}$. *For all* $2 \leq j \leq M$, $\eta_j \Pr[X_t^{(2^i)} > \eta_{j-1}] \leq \frac{\xi_i}{M}$.

**Claim 2.** *Given $i$, $t$, Eq. (3) holds if Condition 2 is satisfied.*

*Proof.* The reason behind the choice of $M$ is to ensure that $\eta_M > 1/(1-\gamma)$ [1]. It follows that, assuming Condition 2 holds, for $1 \leq j \leq M$,

$$E\left[X_t^{(2^i)}\right] = \int_0^{1/(1-\gamma)} \Pr\left[X_t^{(2^i)} > x\right] dx \leq \eta_1 + \sum_{j=2}^M \eta_j \Pr[X_t^{(2^i)} > \eta_{j-1}] \leq \xi_i.$$

$\square$

Therefore, if a time step $t$ is not $\epsilon_2$-optimal, there exists $0 \leq i < \lfloor \log_2 R \rfloor$ and $2 \leq j \leq M$ such that

$$\eta_j \Pr[X_t^{(2^i)} > \eta_{j-1}] > \frac{\xi_i}{M}. \tag{4}$$

Now, the sample complexity can be bounded by the number of $(t, i, j)$ pairs that Eq. (4) is violated. Following the approach of Strehl et al. (2006), for a fixed $(i, j)$-pair, instead of directly counting the number of time steps $t$ such that $\Pr[X_t^{(2^i)} > \eta_{j-1}] > \frac{\xi_i}{M\eta_j}$, we count the number of time steps that $X_t^{(2^i)} > \eta_{j-1}$. Lemma 1 provides an upper bound of the number of such $t$.

### 3.4 KEY LEMMAS

In this section, we present two key lemmas. Lemma 1 bounds the number of sub-optimal actions, which in turn, bounds the sample complexity of our algorithm. Lemma 2 bounds the weighted sum of learning error, i.e. $(\hat{Q}_t - Q^*)(s, a)$, with the sum and maximum of weights. Then, we show that Lemma 1 follows from Lemma 2.

**Lemma 1.** *For fixed $t$ and $\eta > 0$, let $B_\eta^{(t)}$ be the event that $V^*(s_t) - Q^*(s_t, a_t) > \frac{\eta}{1-\gamma}$ in step $t$. If $\eta > 2\epsilon_1$, then with probability at least $1 - \delta/2$,*

$$\sum_{t=1}^{t=\infty} I\left[B_\eta^{(t)}\right] \leq \frac{SA \ln SA \ln 1/\delta}{\eta^2(1-\gamma)^3} \cdot polylog\left(\frac{1}{\epsilon_1}, \frac{1}{1-\gamma}\right), \tag{5}$$

*where $I[\cdot]$ is the indicator function.*

Before presenting Lemma 2, we define a class of sequence that occurs in the proof.

**Definition 3.** *A sequence $(w_t)_{t\geq 1}$ is said to be a $(C, w)$-sequence for $C, w > 0$, if $0 \leq w_t \leq w$ for all $t \geq 1$, and $\sum_{t\geq 1} w_t \leq C$.*

**Lemma 2.** *For every $(C, w)$-sequence $(w_t)_{t\geq 1}$, with probability $1 - \delta/2$, the following holds:*

$$\sum_{t\geq 1} w_t(\hat{Q}_t - Q^*)(s_t, a_t) \leq \frac{C\epsilon_1}{1-\gamma} + \mathcal{O}\left(\frac{\sqrt{wSAC\ell(C)}}{(1-\gamma)^{2.5}} + \frac{wSA \ln C}{(1-\gamma)^3} \ln \frac{1}{(1-\gamma)\epsilon_1}\right).$$

*where $\ell(C) = \iota(C) \ln \frac{1}{(1-\gamma)\epsilon_1}$ is a log-factor.*

Proof of Lemma 2 is quite technical, and is therefore deferred to supplementary materials.

---

[1] $\eta_M > 1/(1-\gamma)$ can be verified by combining inequalities $\xi_i \cdot 2^{M/2} \geq 1/(1-\gamma)$ and $2^{M/2-1} > (M+1)$ for large enough $M$.

Now, we briefly explain how to prove Lemma 1 with Lemma 2. (Full proof can be found in supplementary materials.) Note that since $\hat{Q}_t \geq Q^*$ and $a_t = \arg\max_a \hat{Q}_t(s_t, a)$,

$$V^*(s_t) - Q^*(s_t, a_t) \leq \hat{Q}_t(s_t, a_t) - Q^*(s_t, a_t).$$

We now consider a set $J = \{t : V^*(s_t) - Q^*(s_t, a_t) > \eta(1 - \gamma)^{-1}\}$, and consider the $(|J|, 1)$-weight sequence defined by $w_t = I[t \in J]$. We can now apply Lemma 2 to weighted sum $\sum_{t \geq 1} w_t [V^*(s_t) - Q^*(s_t, a_t)]$. On the one hand, this quantity is obviously at least $|J|\eta(1 - \gamma)^{-1}$. On the other hand, by lemma 2, it is upper bounded by the weighted sum of $(\hat{Q} - Q^*)(s_t, a_t)$. Thus we get

$$|J|\eta(1 - \gamma)^{-1} \leq \frac{C\epsilon_1}{1 - \gamma} + \mathcal{O}\left(\frac{\sqrt{SA|J|\ell(|J|)}}{(1 - \gamma)^{2.5}} + \frac{wSA \ln |J|}{(1 - \gamma)^3} \ln \frac{1}{(1 - \gamma)\epsilon_1}\right).$$

Now focus on the dependence on $|J|$. The left-hand-side has linear dependence on $|J|$, whereas the left-hand-side has a $\tilde{\mathcal{O}}\left(\sqrt{|J|}\right)$ dependence. This allows us to solve out an upper bound on $|J|$ with quadratic dependence on $1/\eta$.

### 3.5 PROOF FOR THEOREM 1

We prove the theorem by stitching Lemma 1 and Condition 2.

*Proof.* (Proof for Theorem 1)

By lemma 1, for any $2 \leq j \leq M$, $\sum_{t=1}^{\infty} I[V^*(s_t) - Q^*(s_t, a_t) > \eta_{j-1}] \leq C$, where

$$C = \frac{SA \ln SA \ln 1/\delta}{\eta_{j-1}^2 (1 - \gamma)^5} \cdot \tilde{P}. \tag{6}$$

Here $\tilde{P}$ is a shorthand for polylog $\left(\frac{1}{\epsilon_1}, \frac{1}{1-\gamma}\right)$.

Let $A_t = I[X_t^{(2^i)} \geq \eta_{j-1}]$ be a Bernoulli random variable, and $\{\mathcal{F}_t\}_{t \geq 1}$ be the filtration generated by random variables $\{(s_\tau, a_\tau) : 1 \leq \tau \leq t\}$. Since $A_t$ is $\mathcal{F}_{t+R}$−measurable, for any $0 \leq k < R$, $\{A_{k+tR} - E[A_{k+tR} \mid \mathcal{F}_{k+tR}]\}_{t \geq 0}$ is a martingale difference sequence. For now, consider a fixed $0 \leq k < R$. By Azuma-Hoeffiding inequality, after $T = \mathcal{O}\left(\frac{C}{2^i} \cdot \frac{M\eta_j}{\xi_i} \ln(RML)\right)$ time steps (if it happens that many times) with

$$\Pr\left[X_{k+tR}^{(2^i)} \geq \eta_{j-1}\right] = \mathbb{E}[A_{k+tR}] > \frac{\xi_i}{M\eta_j}, \tag{7}$$

we have $\sum_t A_{k+tR} \geq C/2^i$ with probability at least $1 - \delta/(2MRL)$.

On the other hand, if $A_{k+tR}$ happens, within $[k + tR, k + tR + R - 1]$, there must be at least $2^i$ time steps at which $V^*(s_t) - Q^*(s_t, a_t) > \eta_{j-1}$. The latter event happens at most $C$ times, and $[k + tR, k + tR + R - 1]$ are disjoint. Therefore, $\sum_{t=0}^{\infty} A_{k+tR} \leq C/2^i$. This suggests that the event described by (7) happens at most $T$ times for fixed $i$ and $j$. Via a union bound on $0 \leq k < R$, we can show that with probability $1 - \delta/(2ML)$, there are at most $RT$ time steps where $\Pr\left[X_t^{(2^i)} \geq \eta_{j-1}\right] > \xi_i/(M\eta_j)$. Thus, the number of sub-optimal steps is bounded by,

$$\sum_{t=1}^{\infty} I[V^*(s_t) - V^{\pi_t}(s_t) > \epsilon]$$

$$\leq \sum_{t=1}^{\infty} \sum_{i=0}^{L} \sum_{j=2}^{M} I\left[\eta_j \Pr[X_t^{(2^i)} > \eta_{j-1}] > \frac{\xi_i}{M}\right] = \sum_{i=0}^{L} \sum_{j=2}^{M} \sum_{t=1}^{\infty} I\left[\Pr[X_t^{(2^i)} > \eta_{j-1}] > \frac{\xi_i}{\eta_j M}\right]$$

$$\leq \sum_{i=0}^{L} \sum_{j=2}^{M} \frac{SAMR \ln 1/\delta \ln SA}{\eta_j \xi_i \cdot 2^i (1 - \gamma)^5} \tilde{P} \leq \sum_{i=0}^{L} \frac{SA \cdot 2^{i+4} \ln SA \ln 1/\delta}{\epsilon_2^2 (1 - \gamma)^6} \tilde{P} \quad \text{(By definition of } \xi_i \text{ and } \eta_j)$$

$$\leq \frac{SAR \ln SA \ln 1/\delta}{\epsilon_2^2 (1 - \gamma)^6} \tilde{P} \leq \frac{SA \ln SA \ln 1/\delta}{\epsilon_2^2 (1 - \gamma)^7} \tilde{P}. \quad \text{(By definition of } R)$$

It should be stressed that throughout the lines, $\tilde{P}$ is a shorthand for an asymptotic expression, instead of an exact value. Our final choice of $\epsilon_2$ and $\epsilon_1$ are $\epsilon_2 = \frac{\epsilon}{3}$, and $\epsilon_1 = \frac{\epsilon}{24RM \ln \frac{1}{1-\gamma}}$. It is not hard to see that $\ln 1/\epsilon_1 = \text{poly}(\ln \frac{1}{\epsilon}, \ln \frac{1}{1-\gamma})$. This immediately implies that with probability $1 - \delta$, the number of time steps such that $(V^* - V^\pi)(s_t) > \epsilon$ is

$$\tilde{\mathcal{O}} \left( \frac{SA \ln 1/\delta}{\epsilon^2 (1-\gamma)^7} \right),$$

where hidden factors are $\text{poly}(\ln \frac{1}{\epsilon}, \ln \frac{1}{1-\gamma}, \ln SA)$. $\square$

## 4 DISCUSSION

In this section, we discuss the implication of our results, and present some interesting properties of our algorithm beyond its sample complexity bound.

### 4.1 COMPARISON WITH PREVIOUS RESULTS

**Lower bound** To the best of our knowledge, the current best lower bound for worst-case sample complexity is $\Omega \left( \frac{SA}{\epsilon^2 (1-\gamma)^3} \ln 1/\delta \right)$ due to Lattimore & Hutter (2012). The gap between our results and this lower bound lies only in the dependence on $1/(1-\gamma)$ and logarithmic terms of $SA$, $1/(1-\gamma)$ and $1/\epsilon$.

**Model-free algorithms** Previously, the best sample complexity bound for a model-free algorithm is $\tilde{\mathcal{O}} \left( \frac{SA}{\epsilon^4 (1-\gamma)^8} \right)$ (suppressing all logarithmic terms), achieved by Delayed Q-learning Strehl et al. (2006). Our results improve this upper bound by a factor of $\frac{1}{\epsilon^2 (1-\gamma)}$, and closes the quadratic gap in $1/\epsilon$ between Delayed Q-learning's result and the lower bound. In fact, the following theorem shows that UCB Q-learning can indeed outperform Delayed Q-learning.

**Theorem 2.** *There exists a family of MDPs with constant $S$ and $A$, in which with probability $1 - \delta$, Delayed Q-learning incurs sample complexity of exploration of $\Omega \left( \frac{\epsilon^{-3}}{\ln(1/\delta)} \right)$, assuming that $\ln(1/\delta) < \epsilon^{-2}$.*

The construction of this hard MDP family is given in the supplementary material.

**Model-based algorithms** For model-based algorithms, better sample complexity results in infinite horizon settings have been claimed Szita & Szepesvári (2010). To the best of our knowledge, the best published result without further restrictions on MDPs is $\tilde{\mathcal{O}} \left( \frac{SA}{\epsilon^2 (1-\gamma)^6} \right)$ claimed by Szita & Szepesvári (2010), which is $(1 - \gamma)$ smaller than our upper bound. From the space complexity point of view, our algorithm is much more memory-efficient. Our algorithm stores $O(SA)$ values, whereas the algorithm in Szita & Szepesvári (2010) needs $\Omega(S^2 A)$ memory to store the transition model.

### 4.2 EXTENSION TO OTHER SETTINGS

Due to length limits, detailed discussion in this section is deferred to supplementary materials.

**Finite horizon MDP** The sample complexity of exploration bounds of UCB Q-learning implies $\tilde{\mathcal{O}}(\epsilon^{-2})$ PAC sample complexity and a $\tilde{\mathcal{O}}(T^{1/2})$ regret bound in finite horizon MDPs. That is, our algorithm implies a PAC algorithm for finite horizon MDPs. We are not aware of reductions of the opposite direction (from finite horizon sample complexity to infinite horizon sample complexity of exploration).

**Regret** The reason why our results can imply an $\tilde{\mathcal{O}}(\sqrt{T})$ regret is that, after choosing $\epsilon_1$, it follows from the argument of Theorem 1 that with probability $1 - \delta$, for all $\epsilon_2 > \tilde{\mathcal{O}}(\epsilon_1/(1-\gamma))$, the number of $\epsilon_2$-suboptimal steps is bounded by

$$\mathcal{O} \left( \frac{SA \ln SA \ln 1/\delta}{\epsilon_2^2 (1-\gamma)^7} \text{polylog} \left( \frac{1}{\epsilon_1}, \frac{1}{1-\gamma} \right) \right).$$

In contrast, Delayed Q-learning Strehl et al. (2006) can only give an upper bound on $\epsilon_1$-suboptimal steps after setting parameter $\epsilon_1$.

## 5 CONCLUSION

Infinite-horizon MDP with discounted reward is a setting that is arguably more difficult than other popular settings, such as finite-horizon MDP. Previously, the best sample complexity bound achieved by model-free reinforcement learning algorithms in this setting is $\tilde{O}(\frac{SA}{\epsilon^4(1-\gamma)^8})$, due to Delayed Q-learning Strehl et al. (2006). In this paper, we propose a variant of Q-learning that incorporates upper confidence bound, and show that it has a sample complexity of $\tilde{\mathcal{O}}(\frac{SA}{\epsilon^2(1-\gamma)^7})$. This matches the best lower bound except in dependence on $1/(1-\gamma)$ and logarithmic factors.

## 6 ACKNOWLEDGEMENTS

The authors thank Chi Jin and Chongjie Zhang for helpful discussions. This work is supported by National Basic Research Program of China (973 Program) (grant no. 2015CB352502), NSFC (61573026), BJNSF (L172037) and Beijing Acedemy of Artificial Intelligence.

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

## A   Proof of Lemma 1

**Lemma 1.** *For fixed $t$ and $\eta > 0$, let $B_\eta^{(t)}$ be the event that $V^*(s_t) - Q^*(s_t, a_t) > \frac{\eta}{1-\gamma}$ in step $t$. If $\eta > 2\epsilon_1$, then with probability at least $1 - \delta/2$,*

$$\sum_{t=1}^{t=\infty} I\left[B_\eta^{(t)}\right] \leq \frac{SA \ln SA \ln 1/\delta}{\eta^2(1-\gamma)^3} \cdot polylog\left(\frac{1}{\epsilon_1}, \frac{1}{1-\gamma}\right), \tag{8}$$

*where $I[\cdot]$ is the indicator function.*

*Proof.* When $\eta > 1$ the lemma holds trivially. Now consider the case that $\eta \leq 1$.

Let $I = \{t : V^*(s_t) - Q^*(s_t, a_t) > \frac{\eta}{1-\gamma}\}$. By lemma 2, with probability $1 - \delta$,

$$\begin{aligned}
\frac{\eta|I|}{1-\gamma} &\leq \sum_{t\in I}(V^*(s_t) - Q^*(s_t, a_t)) \leq \sum_{t\in I}\left[\left(\hat{Q}_t - Q^*\right)(s_t, a_t)\right] \\
&\leq \frac{|I|\epsilon_1}{1-\gamma} + \mathcal{O}\left(\frac{1}{(1-\gamma)^{5/2}}\sqrt{SA|I|\ell(|I|)} + \frac{SA}{(1-\gamma)^3}\ln|I|\ln\frac{1}{\epsilon_1(1-\gamma)}\right) \\
&\leq \frac{|I|\epsilon_1}{1-\gamma} + \mathcal{O}\left(\ln\frac{1}{\epsilon_1(1-\gamma)} \cdot \left(\frac{\sqrt{SA|I|\ln\frac{SA|I|}{\delta}}}{(1-\gamma)^{5/2}} + \frac{SA\ln|I|}{(1-\gamma)^3}\right)\right) \\
&\leq \frac{|I|\epsilon_1}{1-\gamma} + \mathcal{O}\left(\sqrt{\ln\frac{1}{\delta}}\ln\frac{1}{\epsilon_1(1-\gamma)} \cdot \left(\frac{\sqrt{SA|I|\ln SA|I|}}{(1-\gamma)^{5/2}} + \frac{SA\ln|I|}{(1-\gamma)^3}\right)\right)
\end{aligned}$$

Suppose that $|I| = \frac{SAk^2}{\eta^2(1-\gamma)^3}\ln SA$, for some $k > 1$. Then it follows that for some constant $C_1$,

$$\begin{aligned}
\frac{\eta|I|}{1-\gamma} &= \frac{k^2 SA\ln SA}{(1-\gamma)^4\eta} \leq 2\frac{(\eta-\epsilon_1)|I|}{1-\gamma} \\
&\leq C_1\sqrt{\ln\frac{1}{\delta}}\ln\frac{1}{\epsilon_1(1-\gamma)}\left(\frac{\sqrt{SA|I|\ln(SA|I|)}}{(1-\gamma)^{5/2}} + \frac{SA\ln|I|}{(1-\gamma)^3}\right) \\
&\leq C_1\sqrt{\ln\frac{1}{\delta}}\ln\frac{1}{\epsilon_1(1-\gamma)}\left(\frac{SAk}{\eta(1-\gamma)^4}\sqrt{\ln SA\cdot(\ln SA + \ln|I|)} + \frac{SA\ln|I|}{(1-\gamma)^3}\right).
\end{aligned}$$

Therefore

$$
\begin{aligned}
k^2 \ln(SA) &\leq C_1 \sqrt{\ln \frac{1}{\delta}} \ln \frac{1}{\epsilon_1(1-\gamma)} \left( k \left( \ln SA + \ln |I| \right) + \eta(1-\gamma) \ln |I| \right) \\
&\leq k C_1 \sqrt{\ln \frac{1}{\delta}} \ln \frac{1}{\epsilon_1(1-\gamma)} \cdot \left( \ln SA + 2 \ln |I| \right) \\
&\leq k C_1 \sqrt{\ln \frac{1}{\delta}} \ln \frac{1}{\epsilon_1(1-\gamma)} \cdot \left( 3 \ln SA + 4 \ln k + 6 \ln \frac{1}{\eta(1-\gamma)} \right) \\
&\leq 6 k C_1 \sqrt{\ln \frac{1}{\delta}} \ln^2 \frac{1}{\epsilon_1(1-\gamma)} \left( \ln SA + \ln ek \right).
\end{aligned}
$$

Let $C' = \max\{2, 6C_1 \sqrt{\ln \frac{1}{\delta}} \ln^2 \frac{1}{\epsilon_1(1-\gamma)}\}$. Then

$$
k \leq C'(2 + \ln k). \tag{9}
$$

If $k \geq 10 C' \ln C'$, then

$$
\begin{aligned}
k - C'(2 + \ln k) &\geq 8 C' \ln C' - (2 + \ln 10)C' \\
&\geq 4 C'(2 \ln C' - 4) \geq 0,
\end{aligned}
$$

which means violation of (9). Therefore, since $C' \geq 2$

$$
k \leq 10 C' \ln C' \leq 360 C_1^2 \max\{\ln^4 \frac{1}{\epsilon_1(1-\gamma)}, 20 \ln 2\}. \tag{10}
$$

It immediately follows that

$$
|I| = \frac{SAk^2}{\eta^2(1-\gamma)^3} \ln SA \tag{11}
$$

$$
\leq \frac{SA \ln SA}{\eta^2(1-\gamma)^5} \cdot \ln \frac{1}{\delta} \cdot \mathcal{O}\left( \ln^8 \frac{1}{\epsilon_1(1-\gamma)} \right). \tag{12}
$$

$\qquad\qquad\qquad\qquad\qquad\qquad\qquad\qquad\qquad\qquad\qquad\qquad\qquad\qquad\qquad\qquad\qquad\square$

## B PROOF OF LEMMA 2

**Lemma 2.** *For every $(C, w)$-sequence $(w_t)_{t \geq 1}$, with probability $1 - \delta/2$, the following holds:*

$$
\sum_{t \geq 1} w_t (\hat{Q}_t - Q^*)(s_t, a_t) \leq \frac{C\epsilon_1}{1-\gamma} + \mathcal{O}\left( \frac{\sqrt{wSAC\ell(C)}}{(1-\gamma)^{2.5}} + \frac{wSA \ln C}{(1-\gamma)^3} \ln \frac{1}{(1-\gamma)\epsilon_1} \right).
$$

*where $\ell(C) = \iota(C) \ln \frac{1}{(1-\gamma)\epsilon_1}$ is a log-factor.*

**Fact 1.** *(1) The following statement holds throughout the algorithm,*

$$
\hat{Q}_{p+1}(s, a) \leq Q_{p+1}(s, a).
$$

*(2) For any $p$, there exists $p' \leq p$ such that*

$$
\hat{Q}_{p+1}(s, a) \geq Q_{p'+1}(s, a).
$$

*Proof.* Both properties are results of the update rule at line 11 of Algorithm 1. $\qquad\square$

Before proving lemma 2, we will prove two auxiliary lemmas.

**Lemma 3.** *The following properties hold for $\alpha_t^i$ :*

1. $\sqrt{\frac{1}{t}} \le \sum_{i=1}^{t} \alpha_t^i \sqrt{\frac{1}{i}} \le 2\sqrt{\frac{1}{t}}$ for every $t \ge 1, c > 0$.

2. $\max_{i \in [t]} \alpha_t^i \le \frac{2H}{t}$ and $\sum_{i=1}^{t} (\alpha_t^i)^2 \le \frac{2H}{t}$ for every $t \ge 1$.

3. $\sum_{t=i}^{\infty} \alpha_t^i = 1 + 1/H$, for every $i \ge 1$.

4. $\sqrt{\frac{\iota(t)}{t}} \le \sum_{i=1}^{t} \alpha_t^i \sqrt{\frac{\iota(i)}{i}} \le 2\sqrt{\frac{\iota(t)}{t}}$ where $\iota(t) = \ln(c(t+1)(t+2))$, for every $t \ge 1, c \ge 1$.

*Proof.* Recall that

$$\alpha_t = \frac{H+1}{H+t}, \quad \alpha_t^0 = \prod_{j=1}^{t}(1-\alpha_j), \quad \alpha_t^i = \alpha_i \prod_{j=i+1}^{t}(1-\alpha_j).$$

Properties 1-3 are proven by Jin et al. (2018). Now we prove the last property.

On the one hand,

$$\sum_{i=1}^{t} \alpha_t^i \sqrt{\frac{\iota(i)}{i}} \le \sum_{i=1}^{t} \alpha_t^i \sqrt{\frac{\iota(t)}{i}} \le 2\sqrt{\frac{\iota(t)}{t}},$$

where the last inequality follows from property 1.

The left-hand side is proven by induction on $t$. For the base case, when $t = 1, \alpha_t^t = 1$. For $t \ge 2$, we have $\alpha_t^i = (1-\alpha_t)\alpha_{t-1}^i$ for $1 \le i \le t-1$. It follows that

$$\sum_{i=1}^{t} \alpha_t^i \sqrt{\frac{\iota(i)}{i}} = \alpha_t \sqrt{\frac{\iota(t)}{t}} + (1-\alpha_t) \sum_{i=1}^{t-1} \alpha_{t-1}^i \sqrt{\frac{\iota(i)}{i}} \ge \alpha_t \sqrt{\frac{\iota(t)}{t}} + (1-\alpha_t)\sqrt{\frac{\iota(t-1)}{t-1}}.$$

Since function $f(t) = \iota(t)/t$ is monotonically decreasing for $t \ge 1, c \ge 1$, we have

$$\alpha_t \sqrt{\frac{\iota(t)}{t}} + (1-\alpha_t)\sqrt{\frac{\iota(t-1)}{t-1}} \ge \alpha_t \sqrt{\frac{\iota(t)}{t}} + (1-\alpha_t)\sqrt{\frac{\iota(t)}{t}} \ge \sqrt{\frac{\iota(t)}{t}}.$$

$\square$

**Lemma 4.** *With probability at least $1 - \delta/2$, for all $p \ge 0$ and $(s,a)$-pair,*

$$0 \le (Q_p - Q^*)(s,a) \le \frac{\alpha_t^0}{1-\gamma} + \sum_{i=1}^{t} \gamma \alpha_t^i (\hat{V}_{t_i} - V^*)(s_{t_i+1}) + \beta_t, \tag{13}$$

$$0 \le (\hat{Q}_p - Q^*)(s,a), \tag{14}$$

*where $t = N_p(s,a), t_i = \tau(s,a,i)$ and $\beta_t = c_3\sqrt{H\iota(t)/((1-\gamma)^2 t)}$.*

*Proof.* Recall that

$$\alpha_t^0 = \prod_{j=1}^{t}(1-\alpha_j), \quad \alpha_t^i = \alpha_i \prod_{j=i+1}^{t}(1-\alpha_j).$$

From the update rule, it can be seen that our algorithm maintains the following $Q(s,a)$:

$$Q_p(s,a) = \alpha_t^0 \frac{1}{1-\gamma} + \sum_{i=1}^{t} \alpha_t^i \left[ r(s,a) + b_i + \gamma \hat{V}_{t_i}(s_{t_i+1}) \right].$$

Bellman optimality equation gives:

$$Q^*(s,a) = r(s,a) + \gamma \mathbb{P}V^*(s,a) = \alpha_t^0 Q^*(s,a) + \sum_{i=1}^{t} \alpha_t^i \left[ r(s,a) + \gamma \mathbb{P}V^*(s,a) \right].$$

Subtracting the two equations gives

$$(Q_p - Q^*)(s,a) = \alpha_t^0 \left( \frac{1}{1-\gamma} - Q^*(s,a) \right) + \sum_{i=1}^{t} \alpha_t^i \left[ b_i + \gamma \left( V_{t_i} - V^* \right)(s_{t_i+1}) + \gamma \left( V^*(s_{t_i+1}) - \mathbb{P}V^*(s,a) \right) \right].$$

The identity above holds for arbitrary $p$, $s$ and $a$. Now fix $s \in S$, $a \in A$ and $p \in \mathbb{N}$. Let $t = N_p(s, a)$, $t_i = \tau(s, a, i)$. The $t = 0$ case is trivial; we assume $t \geq 1$ below. Now consider an arbitrary fixed $k$. Define

$$\Delta_i = \left( \alpha_k^i \cdot I[t_i < \infty] \cdot \left( \mathbb{P}V^* - \hat{\mathbb{P}}_{t_i} V^* \right)(s, a) \right)$$

Let $F_i$ be the $\sigma$-Field generated by random variables $(s_1, a_1, ..., s_{t_i}, a_{t_i})$. It can be seen that $\mathbb{E}[\Delta_i | F_i] = 0$, while $\Delta_i$ is measurable in $F_{i+1}$. Also, since $0 \leq V^*(s, a) \leq \frac{1}{1-\gamma}$, $|\Delta_i| \leq \frac{2}{1-\gamma}$. Therefore, $\Delta_i$ is a martingale difference sequence; by the Azuma-Hoeffding inequality,

$$\Pr\left[ \left| \sum_{i=1}^k \Delta_i \right| > \eta \right] \leq 2 \exp \left\{ -\frac{\eta^2}{8(1-\gamma)^{-2} \sum_{i=1}^k (\alpha_k^i)^2} \right\}. \tag{15}$$

By choosing $\eta$, we can show that with probability $1 - \delta / [SA(k+1)(k+2)]$,

$$\left| \sum_{i=1}^k \Delta_i \right| \leq \frac{2\sqrt{2}}{1-\gamma} \cdot \sqrt{\sum_{i=1}^k (\alpha_k^i)^2 \cdot \ln \frac{2(k+1)(k+2)SA}{\delta}} \leq \frac{c_2}{1-\gamma} \sqrt{\frac{H\iota(k)}{k}}. \tag{16}$$

Here $c_2 = 4\sqrt{2}$, $\iota(k) = \ln \frac{(k+1)(k+2)SA}{\delta}$. By a union bound for all $k$, this holds for arbitrary $k > 0$, arbitrary $s \in S$, $a \in A$ simultaneously with probability

$$1 - \sum_{s' \in S, a' \in A} \sum_{k=1}^{\infty} \frac{\delta}{2SA(k+1)(k+2)} = 1 - \frac{\delta}{2}.$$

Therefore, we conclude that (16) holds for the random variable $t = N_p(s, a)$ and for all $p$, with probability $1 - \delta/2$ as well.

**Proof of the right hand side of (13):** We also know that $(b_k = \frac{c_2}{1-\gamma} \sqrt{\frac{H\iota(k)}{k}})$

$$\frac{c_2}{1-\gamma} \sqrt{\frac{H\iota(k)}{k}} \leq \sum_{i=1}^k \alpha_k^i b_i \leq \frac{2c_2}{1-\gamma} \sqrt{\frac{H\iota(k)}{k}}.$$

It is implied by (16) that

$$(Q_p - Q^*)(s, a) \leq \frac{\alpha_t^0}{1-\gamma} + \gamma \left| \sum_{i=1}^t \Delta_i \right| + \sum_{i=1}^t \alpha_t^i \left[ \gamma(\hat{V}_{t_i} - V^*)(x_{t_i+1}) + b_i \right]$$

$$\leq \frac{\alpha_t^0}{1-\gamma} + \frac{3c_2}{1-\gamma} \sqrt{\frac{H\iota(t)}{t}} + \sum_{i=1}^t \gamma \alpha_t^i (\hat{V}^{t_i} - V^*)(x_{t_i+1})$$

(Property 4 of lemma 3)

$$\leq \frac{\alpha_t^0}{1-\gamma} + \sum_{i=1}^t \gamma \alpha_t^i (\hat{V}^{t_i} - V^*)(x_{t_i+1}) + \beta_t.$$

Note that $\beta_t = c_3(1-\gamma)^{-1} \sqrt{H\iota(t)/t}$; $c_3 = 3c_2 = 12\sqrt{2}$.

**Proof of the left hand side of (13):** Now, we assume that event that (16) holds. We assert that $Q_p \geq Q^*$ for all $(s, a)$ and $p \leq p'$. This assertion is obviously true when $p' = 0$. Then

$$(Q_p - Q^*)(s, a) \geq -\gamma \left| \sum_{i=1}^t \Delta_i \right| + \sum_{i=1}^t \alpha_t^i \left[ \gamma(\hat{V}_{t_i} - V^*)(x_{t_i+1}) + b_i \right]$$

$$\geq \sum_{i=1}^t \alpha_t^i b_i - \gamma \left| \sum_{i=1}^t \Delta_i \right| \geq 0.$$

Therefore the assertion holds for $p' + 1$ as well. By induction, it holds for all $p$.

We now see that (13) holds for probability $1 - \delta/2$ for all $p$, $s$, $a$. Since $\hat{Q}_p(s, a)$ is always greater than $Q_{p'}(s, a)$ for some $p' \leq p$, we know that $\hat{Q}_p(s, a) \geq Q_{p'}(s, a) \geq Q^*(s, a)$, thus proving (14).

□

We now give a proof for lemma 2. Recall the definition for a $(C, w)$-sequence. A sequence $(w_t)_{t \geq 1}$ is said to be a $(C, w)$-sequence for $C, w > 0$, if $0 \leq w_t \leq w$ for all $t \geq 1$, and $\sum_{t \geq 1} w_t \leq C$.

*Proof.* Let $n_t = N_t(s_t, a_t)$ for simplicity; we have

$$\sum_{t \geq 1} w_t (\hat{Q}_t - Q^*)(s_t, a_t)$$

$$\leq \sum_{t \geq 1} w_t (Q_t - Q^*)(s_t, a_t)$$

$$\leq \sum_{t \geq 1} w_t \left[ \frac{\alpha_{n_t}^0}{1 - \gamma} + \beta_{n_t} + \gamma \sum_{i=1}^{n_t} \alpha_{n_t}^i \left( \hat{V}_{\tau(s_t, a_t, i)} - V^* \right) (s_{\tau(s_t, a_t, i)+1}) \right] \quad (17)$$

The last inequality is due to lemma 4. Note that $\alpha_{n_t}^0 = \mathbb{I}[n_t = 0]$, the first term in the summation can be bounded by,

$$\sum_{t \geq 1} w_t \frac{\alpha_{n_t}^0}{1 - \gamma} \leq \frac{SAw}{1 - \gamma}. \quad (18)$$

For the second term, define $u(s, a) = \sup_t N_t(s, a)$.[2] It follows that,

$$\sum_{t \geq 1} w_t \beta_{n_t} = \sum_{s,a} \sum_{i=1}^{u(s,a)} w_{\tau(s,a,i)} \beta_i$$

$$\leq \sum_{s,a} (1 - \gamma)^{-1} c_3 \sum_{i=1}^{C_{s,a}/w} \sqrt{\frac{H\iota(i)}{i}} w \quad (19)$$

$$\leq 2 \sum_{s,a} (1 - \gamma)^{-1} c_3 \sqrt{\iota(C) H C_{s,a} w} \quad (20)$$

$$\leq 2 c_3 (1 - \gamma)^{-1} \sqrt{w SAH C \iota(C)}. \quad (21)$$

Where $C_{s,a} = \sum_{t \geq 1, (s_t, a_t) = (s,a)} w_t$. Inequality (19) follows from rearrangement inequality, since $\iota(x)/x$ is monotonically decreasing. Inequality (21) follows from Jensen's inequality.

For the third term of the summation, we have

$$\sum_{t \geq 1} w_t \sum_{i=1}^{n_t} \alpha_{n_t}^i \left( \hat{V}_{\tau(s_t, a_t, i)} - V^* \right) (s_{\tau(s_t, a_t, i)+1})$$

$$\leq \sum_{t' \geq 1} \left( \hat{V}_{t'} - V^* \right) (s_{t'+1}) \left( \sum_{\substack{t = t'+1 \\ (s_t, a_t) = (s'_t, a'_t)}}^{\infty} \alpha_{n_t}^{n_{t'}} w_t \right). \quad (22)$$

$$(23)$$

Define

$$w'_{t'+1} = \left( \sum_{\substack{t = t'+1 \\ (s_t, a_t) = (s'_t, a'_t)}}^{\infty} \alpha_{n_t}^{n_{t'}} w_t \right).$$

We claim that $w'_{t'+1}$ is a $(C, (1 + \frac{1}{H})w)$-sequence. We now prove this claim. By lemma 3, for any $t' \geq 0$,

$$w'_{t'+1} \leq w \sum_{j=n_{t'}}^{\infty} \alpha_j^{n_{t'}} = (1 + 1/H)w.$$

---

[2] $u(s, a)$ could be infinity when $(s, a)$ is visited for infinite number of times.

By $\sum_{j=0}^{i} \alpha_i^j = 1$, we have $\sum_{t' \geq 1} w'_{t'+1} \leq \sum_{t \geq 1} w_t \leq C$. This proves the assertion. It follows from (22) that

$$\sum_{t \geq 1} w'_{t+1} \left( \hat{V}_t - V^* \right) (s_{t+1})$$

$$= \sum_{t \geq 1} w'_{t+1} \left( \hat{V}_{t+1} - V^* \right) (s_{t+1}) + \sum_{t \geq 1} w'_{t+1} \left( \hat{V}_t - \hat{V}_{t+1} \right) (s_{t+1}) \tag{24}$$

$$\leq \sum_{t \geq 1} w'_{t+1} \left( \hat{V}_{t+1} - V^* \right) (s_{t+1}) + \sum_{t \geq 1} w'_{t+1} \left( 2\alpha_{n_t+1} \frac{1}{1-\gamma} \right) \tag{25}$$

$$\leq \sum_{t \geq 1} w'_{t+1} \left( \hat{V}_{t+1} - V^* \right) (s_{t+1}) + \mathcal{O} \left( \frac{wSAH}{1-\gamma} \ln C \right) \tag{26}$$

$$\leq \sum_{t \geq 1} w'_{t+1} \left( \hat{Q}_{t+1} - Q^* \right) (s_{t+1}, a_{t+1}) + \mathcal{O} \left( \frac{wSAH}{1-\gamma} \ln C \right) \tag{27}$$

Inequality (25) comes from the update rule of our algorithm. Inequality (26) comes from the fact that $\alpha_t = (H+1)/(H+t) \leq H/t$ and Jensen's Inequality. More specifically, let $C'_{s,a} = \sum_{t \geq 1, (s_t, a_t = s,a)} w'_{t+1}$, $w' = w(1 + 1/H)$. Then

$$\sum_{t \geq 1} w'_{t+1} \alpha_{n_t+1} \leq \sum_{s,a} \sum_{n=1}^{C'_{s,a}/w'} w' \frac{H}{n} \leq \sum_{s,a} Hw' \ln(C'_{s,a}/w) \leq 2SAHw \ln C.$$

Putting (18), (21) and (27) together, we have,

$$\sum_{t \geq 1} w_t (\hat{Q}_t - Q^*)(s_t, a_t)$$

$$\leq 2c_3 \frac{\sqrt{wSAHC\iota(C)}}{1-\gamma} + \mathcal{O} \left( \frac{wSAH}{1-\gamma} \ln C \right) + \gamma \sum_{t \geq 1} w'_{t+1} \left( \hat{Q}_{t+1} - Q^* \right) (s_{t+1}, a_{t+1}). \tag{28}$$

Observe that the third term is another weighted sum with the same form as (17). Therefore, we can unroll this term repetitively with changing weight sequences. Suppose that our original weight sequence is also denoted by $\{w_t^{(0)}\}_{t \geq 1}$, while $\{w_t^{(k)}\}_{t \geq 1}$ denotes the weight sequence after unrolling for $k$ times. Let $w^{(k)}$ be $w \cdot (1 + 1/H)^k$. Then we can see that $\{w_t^{(k)}\}_{t \geq 1}$ is a $(C, w^{(k)})$-sequence. Suppose that we unroll for $H$ times. Then

$$\sum_{t \geq 1} w_t (\hat{Q}_t - Q^*)(s_t, a_t)$$

$$\leq 2c_3 \frac{\sqrt{w^{(H)} SAHC\iota(C)}}{(1-\gamma)^2} + \mathcal{O} \left( \frac{w^{(H)} SAH}{(1-\gamma)^2} \ln C \right) + \gamma^H \sum_{t \geq 1} w_t^{(H)} \left( \hat{Q}_t - Q^* \right) (s_t, a_t)$$

$$\leq 2c_3 \frac{\sqrt{w^{(H)} SAHC\iota(C)}}{(1-\gamma)^2} + \mathcal{O} \left( \frac{w^{(H)} SAH}{(1-\gamma)^2} \ln C \right) + \gamma^H \frac{C}{1-\gamma}.$$

We set $H = \frac{\ln 1/((1-\gamma)\epsilon_1)}{\ln 1/\gamma} \leq \frac{\ln 1/((1-\gamma)\epsilon_1)}{1-\gamma}$. It follows that $w^{(H)} = (1 + 1/H)^H w^{(0)} \leq e w^{(0)}$, and that $\gamma^H \frac{C}{1-\gamma} \leq C\epsilon_1$. Also, let $\ell(C) = \iota(C) \ln((1-\gamma)^{-1} \epsilon_1^{-1})$. Therefore,

$$\sum_{t \geq 1} w_t (\hat{Q}_t - Q^*)(s_t, a_t) \leq \frac{C\epsilon_1}{1-\gamma} + \mathcal{O} \left( \frac{\sqrt{wSAC\ell(C)}}{(1-\gamma)^{2.5}} + \frac{wSA}{(1-\gamma)^3} \ln C \ln \frac{1}{(1-\gamma)\epsilon_1} \right). \tag{29}$$

$$\square$$

## C    EXTENSION TO OTHER SETTINGS

First we define a mapping from a finite horizon MDP to an infinite horizon MDP so that our algorithm can be applied. For an arbitrary finite horizon MDP $\mathcal{M} = (S, A, H, r_h(s, a), p_h(s' \mid s, a))$ where $H$ is the length of episode, the corresponding infinite horizon MDP $\bar{\mathcal{M}} = (\bar{S}, \bar{A}, \gamma, \bar{r}(\bar{s}, \bar{a}), \bar{p}(\bar{s}' \mid \bar{s}, \bar{a}))$ is defined as,

- $\bar{S} = S \times H, \bar{A} = A$;
- $\gamma = (1 - 1/H)$;
- for a state $s$ at step $h$, let $\bar{s}_{s,h}$ be the corresponding state. For any action $a$ and next state $s'$, define $\bar{r}(\bar{s}_{s,h}, a) = \gamma^{H-h+1} r_h(s, a)$ and $\bar{p}(\bar{s}_{s',h+1} \mid \bar{s}_{s,h}, a) = p_h(s' \mid s, h)$. And for $h = H$, set $\bar{r}(\bar{s}_{s,h}, a) = 0$ and $\bar{p}(\bar{s}_{s',1} \mid \bar{s}_{s,h}, a) = I[s' = s_1]$ for a fixed starting state $s_1$.

Let $\bar{V}_t$ be the value function in $\bar{\mathcal{M}}$ at time $t$ and $V_h^k$ the value function in $\mathcal{M}$ at episode $k$, step $h$. It follows that $\bar{V}^*(\bar{s}_{s_1,1}) = \frac{\gamma^H}{1-\gamma^H} V_1^*(s_1)$. And the policy mapping is defined as $\pi_h(s) = \bar{\pi}(\bar{s}_{s,h})$ for policy $\bar{\pi}$ in $\bar{\mathcal{M}}$. Value functions in MDP $\mathcal{M}$ and $\bar{\mathcal{M}}$ are closely related in a sense that, any $\epsilon$-optimal policy $\bar{\pi}$ of $\bar{\mathcal{M}}$ corresponding to an $(\epsilon/\gamma^H)$-optimal policy $\pi$ in $\mathcal{M}$ (see section C.1 for proof). Note that here $\gamma^H = (1 - 1/H)^H = \mathcal{O}(1)$ is a constant.

For any $\epsilon > 0$, by running our algorithm on $\bar{M}$ for $\tilde{\mathcal{O}}(\frac{3SAH^9}{\epsilon^2})$ time steps, the starting state $s_1$ is visited at least $\tilde{\mathcal{O}}(\frac{3SAH^8}{\epsilon^2})$ times, and at most $1/3$ of them are not $\epsilon$-optimal. If we select the policy uniformly randomly from the policy $\pi^{tH+1}$ for $0 \leq t < T/H$, with probability at least $2/3$ we can get an $\epsilon$-optimal policy. Therefore the PAC sample complexity is $\tilde{\mathcal{O}}\left(\epsilon^{-2}\right)$ after hiding $S, A, H$ terms.

On the other hand, we want to show that for any $K$ episodes,

$$\text{Regret}(T) = \sum_{k=1}^{T/H} \left[V^*(s_1) - V_1^k(s_1)\right] \propto T^{1/2}.$$

The reason why our algorithm can have a better reduction from regret to PAC is that, after choosing $\epsilon_1$, it follows from the argument of theorem 1 that for all $\epsilon_2 > \tilde{\mathcal{O}}(\epsilon_1/(1 - \gamma))$, the number of $\epsilon_2$-suboptimal steps is bounded by

$$\mathcal{O}\left(\frac{SA \ln SA \ln 1/\delta}{\epsilon_2^2 (1 - \gamma)^7} \text{polylog}\left(\frac{1}{\epsilon_1}, \frac{1}{1 - \gamma}\right)\right)$$

with probability $1 - \delta$. In contrast, delayed Q-learning can only give an upper bound on $\epsilon_1$-suboptimal steps after setting parameter $\epsilon_1$.

Formally, let $X_k = V^*(s_1) - V_1^k(s_1)$ be the regret of $k$-th episode. For any $T$, set $\epsilon = \sqrt{SA/T}$ and $\epsilon_2 = \tilde{\mathcal{O}}(\epsilon_1/(1 - \gamma))$. Let $M = \lceil \log_2 \frac{1}{\epsilon_2(1-\gamma)} \rceil$. It follows that,

$$\text{Regret}(T) \leq T\epsilon_2 + \sum_{i=1}^{M} \left(\left|k : \{X_k \geq \epsilon_2 \cdot 2^{i-1}\}\right|\right) \epsilon_2 \cdot 2^i$$

$$\leq \tilde{\mathcal{O}}\left(T\epsilon_2 + \sum_{i=1}^{M} \frac{SA \ln 1/\delta}{\epsilon_2 \cdot 2^{i-2}}\right)$$

$$\leq \tilde{\mathcal{O}}\left(\sqrt{SAT} \ln 1/\delta\right)$$

with probability $1 - \delta$. Note that the $\tilde{\mathcal{O}}$ notation hides the poly $(1/(1 - \gamma), \log 1/\epsilon_1)$ which is, by our reduction, poly $(H, \log T, \log S, \log A)$.

### C.1    CONNECTION BETWEEN VALUE FUNCTIONS

Recall that our MDP mapping from $\mathcal{M} = (S, A, H, r_h(s, a), p_h(s' \mid s, a))$ to $\bar{\mathcal{M}} = (\bar{S}, \bar{A}, \gamma, \bar{r}(\bar{s}, \bar{a}), \bar{p}(\bar{s}' \mid \bar{s}, \bar{a}))$ is defined as,

- $\bar{S} = S \times H, \bar{A} = A$;
- $\gamma = (1 - 1/H)$;
- for a state $s$ at step $h$, let $\bar{s}_{s,h}$ be the corresponding state. For any action $a$ and next state $s'$, define $\bar{r}(\bar{s}_{s,h}, a) = \gamma^{H-h+1} r_h(s, a)$ and $\bar{p}(\bar{s}_{s',h+1} \mid \bar{s}_{s,h}, a) = p_h(s, h)$. And for $h = H$, set $\bar{r}(\bar{s}_{s,h}, a) = 0$ and $\bar{p}(\bar{s}_{s',1} \mid \bar{s}_{s,h}, a) = I[s' = s_1]$ for a fixed starting state $s_1$.

For a trajectory $\{(\bar{s}_{s_1,1}, \bar{a}_1), (\bar{s}_{s_2,2}, \bar{a}_2), \cdots\}$ in $\bar{\mathcal{M}}$, let $\{(s_1, a_1), (s_2, a_2), \cdots\}$ be the corresponding trajectory in $\mathcal{M}$. Note that $\mathcal{M}$ has a unique fixed starting state $s_1$, which means that $s_{tH+1} = s_1$ for all $t \geq 0$. Denote the corresponding policy of $\bar{\pi}^t$ as $\pi^t$ (may be non-stationary), then we have

$$\bar{V}^{\bar{\pi}^t}(\bar{s}_{s_1,1}) = \mathbb{E}\left[\bar{r}(\bar{s}_{s_1,1}, \bar{a}_1) + \gamma\bar{r}(\bar{s}_{s_2,2}, \bar{a}_2) + \cdots + \gamma^{H-1}\bar{r}(\bar{s}_{s_{H-1},H-1}, \bar{a}_{H-1}) + \gamma^H \bar{V}^{\pi_{t+H-1}}(\bar{s}_{s_{H+1},1})\right]$$
$$= \gamma^H \mathbb{E}\left[r_1(s_1, a_1) + r_2(s_2, a_2) + \cdots + r_{H-1}(s_{H-1}, a_{H-1}) + \bar{V}^{\pi_{t+H}}(\bar{s}_{s_{H+1},1})\right]$$
$$= \gamma^H V^{\pi^t}(s_1) + \gamma^H \bar{V}^{\pi_{t+H}}(\bar{s}_{s_1,1}).$$

Then for a stationary policy $\bar{\pi}$, we can conclude $\bar{V}^{\bar{\pi}}(\bar{s}_{s_1,1}) = \frac{\gamma^H}{1-\gamma^H} V^\pi(s_1)$. Since the optimal policy $\bar{\pi}^*$ is stationary, we have $\bar{V}^*(\bar{s}_{s_1,1}) = \frac{\gamma^H}{1-\gamma^H} V^*(s_1)$.

By definition, $\bar{\pi}$ is $\epsilon$-optimal at time step $t$ means that

$$\bar{V}^{\bar{\pi}^t}(\bar{s}_{s_1,1}) \geq \bar{V}^*(\bar{s}_{s_1,1}) - \epsilon.$$

It follows that

$$\gamma^H V^{\pi^t}(s_1) + \gamma^H \bar{V}^{\pi_{t+H}}(\bar{s}_{s_1,1}) = \bar{V}^{\bar{\pi}}(\bar{s}_{s_1,1}) \geq \bar{V}^*(\bar{s}_{s_1,1}) - \epsilon,$$

hence

$$\gamma^H V^{\pi^t}(s_1) \geq (1-\gamma^H)\bar{V}^*(\bar{s}_{s_1,1}) + \gamma^H(\bar{V}^*(\bar{s}_{s_1,1}) - \bar{V}^{\pi_{t+H}}(\bar{s}_{s_1,1})) - \epsilon \geq (1-\gamma^H)\bar{V}^*(\bar{s}_{s_1,1}) - \epsilon.$$

Therefore we have

$$V^{\pi^t}(s_1) \geq \frac{1-\gamma^H}{\gamma^H}\bar{V}^*(\bar{s}_{s_1,1}) - \epsilon/\gamma^H = V^*(s_1) - \epsilon/\gamma^H,$$

which means that $\pi^t$ is an $(\epsilon/\gamma^H)$-optimal policy.

## D  A HARD INSTANCE FOR DELAYED Q-LEARNING

In this section, we prove Theorem 2 regarding the performance of Delayed Q-learning.

**Theorem 2.** *There exists a family of MDPs with constant $S$ and $A$, in which with probability $1 - \delta$, Delayed Q-learning incurs sample complexity of exploration of $\Omega\left(\frac{\epsilon^{-3}}{\ln(1/\delta)}\right)$, assuming that $\ln(1/\delta) < \epsilon^{-2}$.*

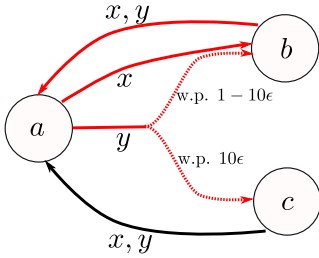

Figure 1: The MDP family. Actions are denoted by arrows. Actions with red color have reward 1, and reward 0 otherwise.

*Proof.* For each $0 < \epsilon < \frac{1}{10}$, consider the following MDP (see also Fig. 1): state space is $\mathcal{S} = \{a, b, c\}$ while action set is $\mathcal{A} = \{x, y\}$; transition probabilities are $P(b|a, y) = 1 - 10\epsilon$, $P(c|a, y) = 10\epsilon$, $P(b|a, x) = 1$, $P(a|b, \cdot) = P(a|c, \cdot) = 1$. Rewards are all 1, except $R(c, \cdot) = 0$.

Assume that Delayed Q-learning is called for this MDP starting from state $a$, with discount $\gamma > \frac{1}{2}$ and precision set as $\epsilon$. Denote the $Q$ value maintained by the algorithm by $\hat{Q}$. Without loss of generality, assume that the initial tie-breaking favors action $y$ when comparing $\hat{Q}(a, x)$ and $\hat{Q}(a, y)$. In that case, unless $\hat{Q}(a, y)$ is updated, the agent will always choose $y$ in state $a$. Since $Q(a, x) - Q(a, y) = 10\epsilon\gamma > \epsilon$ for any policy, choosing $y$ at state $a$ implies that the timestep is not $\epsilon$-optimal. In other words, sample complexity for exploration is at least the number of times the agent visits $a$ before the first update of $\hat{Q}(a, y)$.

In the Delayed Q-learning algorithm, $\hat{Q}(\cdot, \cdot)$ are initialized to $1/(1 - \gamma)$. Therefore, $\hat{Q}(a, y)$ could only be updated if $\max \hat{Q}(c, \cdot)$ is updated (and becomes smaller than $1/(1 - \gamma)$). According to the algorithm, this can only happen if $c$ is visited $m = \Omega\left(\frac{1}{\epsilon^2}\right)$ times.

However, each time the agent visits $a$, there is less than $10\epsilon$ probability of transiting to $c$. Let $t_0 = m/(10\epsilon C)$, where $C = 3\ln\frac{1}{\delta} + 1$. $\delta$ is chosen such that $C \leq m$. In the first $2t_0$ timesteps, $a$ will be visited $t_0$ times. By Chernoff's bound, with probability $1 - \delta$, state $c$ will be visited less than $m$ times. In that case, $\hat{Q}(a, y)$ will not be updated in the first $2t_0$ timesteps. Therefore, with probability $1 - \delta$, sample complexity of exploration is at least

$$t_0 = \Omega\left(\frac{1}{\epsilon^3 \left(\ln 1/\delta\right)}\right).$$

When $\ln(1/\delta) < \epsilon^{-2}$, it can be seen that $C = 3\ln\frac{1}{\delta} + 1 < \frac{4}{\epsilon^2} < m$. $\qquad\square$

