# OpenReview forum: "Q-learning with UCB Exploration is Sample Efficient for Infinite-Horizon MDP"
_ICLR.cc/2020/Conference — Accept (Poster)_

### Official Review · AnonReviewer1 · 2019-10-23
**Official Blind Review #1**

**Rating:** 6

**Review:**

This paper considered a Q-learning algorithm with UCB exploration policy for infinite-horizon MDP, and derived the sample complexity of exploration bound. The bound was shown to improve the existing results and matched the lower bound up to some log factors. The problem considered is interesting and challenging. However, I have a few major concerns.

1. Assumptions: The Condition 2 is on the term $X_t^{(i)}$ which is the i-th largest item of the gap $\Delta_t$ (the difference between the value function and the Q function fo the optimal policy). How to verify this condition in practice? Do you need this condition for all $t$? Does this condition depend on the choice of length $R$? The conditions are listed without any discussions.

2. Algorithm: The Algorithm depends on the choice of the parameter $R$. How to choose $R$ in practice?

3. Writing: This paper is not well written. There are many typos and grammar errors. In addition, the key components Sections 3.3 and 3.4 are very hard to follow. For instance, in Section 3.3, the authors first introduced Condition 1 and then Condition 2, and then claimed that Condition 2 implied Condition 1. Similarly, in Section 3.4, Lemma 1 was introduced before Lemma 2. Then the authors claimed that Lemma 2 implies Lemma 1. If would be easier to follow if the authors only introduced the latter result, and then discuss the former result as a remark.

~~~~~~~~~~~~~~~~~~~~~~
After rebuttal: Thanks the authors for addressing my questions. My first two major concerns have been nicely addressed. Therefore, I would like to increase my rating to 6.


**Experience Assessment:**

I have read many papers in this area.

**Review Assessment: Checking Correctness Of Derivations And Theory:**

I assessed the sensibility of the derivations and theory.

**Review Assessment: Checking Correctness Of Experiments:**

N/A

**Review Assessment: Thoroughness In Paper Reading:**

I made a quick assessment of this paper.

---

> ### Author Response · Authors · 2019-11-10
> **Response to reviewer 1**
>
> We thank anonymous reviewer 1 for the review.
>
> Condition 2 is a sufficient condition for near-optimal reward. Lemma 1 proved that the number of steps violating condition 2 is bounded. Please note that our algorithm achieves finite sample complexity of exploration in any MDPs, regardless of the suboptimality gap.
>
> Once the desired performance of our algorithm (namely, epsilon) is determined, $R$ can be chosen as $R=\ln \left(\frac{3}{\epsilon (1−\gamma)}\right) /(1 − \gamma).$

---

> > ### Comment · AnonReviewer1 · 2019-11-13
> > **my questions were not answered**
> >
> > Thanks the author for addressing my Question 2. However, I do not think my Question 1 has been clearly addressed.
> >
> > Condition2: Do you need this condition for all t? Does this condition depend on the choice of length R? How to verify Condition 2 in practice?

---

> > > ### Author Response · Authors · 2019-11-14
> > > **Thanks for the feedback**
> > >
> > > We thank the reviewer for the feedback.
> > >
> > > First, we would like to clarify that both Condition 1 and Condition 2 are not assumptions for our theorem to hold. We do not have additional assumptions beyond the problem formulation. Instead, they are simply used as a proxy for sample complexity in our proof.
> > >
> > > Below we answer the three questions raised by the reviewer:
> > >
> > > 1.Do we need Condition 2 for all t?
> > > We do not require Condition 2 to hold for all t. Instead, we count the number of times such that Condition 2 is violated, which is an upper bound for sample complexity. This counting is done in the series of inequalities on page 7 (above the final bound), where it can be noted that the second line counts the number of times such that Condition 2 is violated.
> > >
> > > 2.Does this condition depend on the choice of length R?
> > > The condition itself does depend on the choice of $R$, in the sense that the logarithmic of $R$ is involved in the definition of $M$.
> > >
> > > 3.How to verify Condition 2 in practice?
> > > Condition 2 is not an assumption of the theorem. It is a statement to make the roadmap of the proof easier to understand. Verifying whether Condition 2 holds at a particular time step is similar to verifying whether the algorithm is performing bad at a particular time. Currently we think this is hard but unnecessary.

---

### Official Review · AnonReviewer2 · 2019-10-24
**Official Blind Review #2**

**Rating:** 6

**Review:**

Summary:
In this paper, the authors extend the UCB Q-learning algorithm by Jin et al. (2018) to infinite horizon discounted MDPs, and prove a PAC bound of \tilde{O}(SA/\epsilon^2 (1-\gamma)^7) for the resulting algorithm. This bound improves the one for delayed Q-learning by Strehl et al. (2006) and matches the lower-bound in terms of \epsilon, S, and A.


Comments:
- Overall, the paper is well-written and well-structured. Although most of the results are in the appendix, the authors have done a good job in what they reported in the paper and what they left for the appendix.
- I personally think ICLR is not a good venue for this kind of papers. Places like COLT and journals are better because they allow the authors to report most of the results in the main paper, and give more time to the reviewers to go over all the proofs.
- I did not have time to go over all the proofs in the appendix but I checked those in the paper. I did not find any error, but the math can be written more rigorously.
- The algorithm is quite similar to the one by Jin et al. (2018), which is for finite-horizon problems. The authors discuss on Page 4 that why despite the similarity of the algorithms, the techniques in Jin et al. (2018) cannot be directly applied to their case. It would be good if the authors have a discussion on the novelty of the techniques used in their analysis, not only w.r.t. Jin et al. (2018), but w.r.t. other analysis of model free algorithms in infinite-horizon discounted problems.
- What makes this work different than Jin et al. (2018), and is its novelty, is obtaining a PAC bound. The resulting regret bound is similar to the one in Jin et al. (2018), but the PAC bound is new. It would be good if the authors mention this on Page 4. This makes it more clear in which sense this work is different than that by Jin et al. (2018).
- It is important to write in the statement of Condition 2 that "i" changes between 0 and \log_2 R.


**Experience Assessment:**

I have published one or two papers in this area.

**Review Assessment: Checking Correctness Of Derivations And Theory:**

I assessed the sensibility of the derivations and theory.

**Review Assessment: Checking Correctness Of Experiments:**

N/A

**Review Assessment: Thoroughness In Paper Reading:**

I read the paper at least twice and used my best judgement in assessing the paper.

---

> ### Author Response · Authors · 2019-11-10
> **Response  to Reviewer 2**
>
> We thank anonymous reviewer 2 for the review.
>
> Comparison to other previous works: The most notable prior work in model-free PAC-RL is Delayed Q-learning  (Strehl et al. 2006), which has a $\tilde{O}\left(SA/\epsilon^4(1-\gamma)^8\right)$ sample complexity of exploration bound. Our improved dependence on $\epsilon$ comes from two factors.
>
> 1. Strehl et al. treat all mistakes over $\epsilon$ on the same footing. In our analysis, we require large mistakes to happen rarely, but allow small mistakes to happen with larger probabilities. This is mainly captured by Condition 2.
>
> 2. Delayed Q-learning needs $\epsilon^{-2}$ samples to update the Q-value by $\epsilon$, which results in slow learning progress at the start. This is illustrated by our example in Appendix D. In comparison, UCB bonus allows faster progress at the start.

---

### Official Review · AnonReviewer3 · 2019-10-26
**Official Blind Review #3**

**Rating:** 6

**Review:**

This paper extends Jin et al. (2018)'s idea to infinite horizon and improves the best known sample complexity to $\tilde{O}(\frac{SA}{\epsilon^2 (1-\gamma)^7})$. The derivation is similar to Jin's paper except a very careful selection on the pseudo-horizon length $H$, where $H$ is given in finite horizon and work as the decaying rate for $\alpha_k$, but for infinite horizon when we need to decide how to pick $H$.

Theoretical Soundness: I didn't check every step of the proof, but I the steps I check is correct and I can feel that the derivation is solid.

Novelty: In section 3.2, the authors discuss the difference between finite case and infinite case. I don't agree with the example if the starting state $s_1$ is with leaving probability less than $T^{-1}$. In this case, the latter rewards counted for $V(s_1)$ will multiply by $\gamma^t$ which is pretty small, and will not contribute too much on the error. But I do agree that the case of infinite horizon is different since we need to carefully decide the decaying rate of $\alpha_k$, which is definitely related to $\frac{1}{1-\gamma}$ but we need to figure out the relation. I think that is the most difficult part of the whole proof and this is the main technical contribution for the paper.

Minor issue: In algorithm box I didn't see $V^{\pi_t}, \hat{Q}_t$ which occurs a lot in the derivation. Maybe change $\hat{V}$ and $\hat{Q}$ in the way align to the derivation?

Overall I think this paper is sufficient to get in the conference. But to be honest since I lack the background of PAC-RL, I would remain a conservative of weak accept and would like to hear more discussions from other reviewers and authors to finalize my decision.

**Experience Assessment:**

I do not know much about this area.

**Review Assessment: Checking Correctness Of Derivations And Theory:**

I assessed the sensibility of the derivations and theory.

**Review Assessment: Checking Correctness Of Experiments:**

N/A

**Review Assessment: Thoroughness In Paper Reading:**

N/A

---

> ### Author Response · Authors · 2019-11-10
> **Response to Reviewer 3**
>
> We thank anonymous reviewer 3 for the review.
>
> Regarding the example in section 3.2: Please note that ‘sample complexity of exploration’ counts the number of sub-optimal steps. We agree that steps taken in $s_1$ maybe near-optimal. When the agent leaves state $s_1$ and enters other state, it will take sub-optimal steps, which contributes to the sample complexity of exploration.
> For the finite horizon case, when the leaving probability is small, the regret could be small.
>
> In summary, this example showed that there are cases where an algorithm is efficient when using regret as complexity measure and inefficient when using sample complexity of exploration. Therefore there is no straight-forward reduction from infinite horizon case to finite horizon case.

---

### Official Review · AnonReviewer4 · 2019-10-27
**Official Blind Review #4**

**Rating:** 6

**Review:**

Summary: This paper adapts the UCB Q-learning method to the inifinite-horizon discounted MDP setting. With an analysis similar to that of Jin et al (2018), it shows that this algorithm achieves a PAC bound of (1-gamma)^-7 |S||A|/eps^2, improving previous best-known bound (Delayed Q-learning, Strehlet et al, 2006, (1-gamma)^-8 |S||A|/eps^4) for this case.

Evaluation: As I see this paper a direct extension of that of Jin et al (2018), I am afraid I have to recommend a rejection.

Here are some more detailed comments:

Significance:
This paper studies the RL problem for the infinite-horizon discounted MDP setting. This is an important setting in reinforcement learning. However, the bound is not optimal as the dependence of (1-gamma) is significantly larger than the lower bound. Moreover, both the algorithm and analysis are direct extensions of that of Jin et al, I do not see a huge technique improvement.

Technique Novelty:
As stated in the paper, the major difficulty is that the inf-horizon case does not have a set of "consecutive episodes". Therefore the "learning error at time t cannot be decomposed as errors from a set of consecutive time
steps before t, but errors from a set of non-consecutive time steps without any structure." However, I do not see a major technological innovation is needed to get around this issue. As a result, the analysis and algorithm in this paper are very similar to that of Jin et al 2018, who nearly implicitly contain the results in this paper.

Furthermore, I would think there is a (likely) very simple reduction from the inf-horizon to finite-horizon: break the inifinite horizon into episodes of length R = O((1-\gamma)^-1 log(eps^-1)). Now, although the MDP does not restart, but it can be treated as restarting at a history-dependent initial state distribution at the beginning of every episode. So, an optimal finit-horizon algorithm in this setting is at most epsilon worse than the optimal inf-horizon algorithm, no matter where/when you start. With little to no modification, we can see that Jin et al work in this setting. Thus, we obtain an algorithm for the inf-horizon as well.

A good match for this conference?
As this paper is an adaptation of a previously known Q-learning algorithm to a slightly different setting in RL, I do not see how it fits the "learning representation" paradigm. Of course, Q-function can be argued as a representation of the MDP model, but this Q-function itself is not a new concept in this paper.





**Experience Assessment:**

I have published in this field for several years.

**Review Assessment: Checking Correctness Of Derivations And Theory:**

I assessed the sensibility of the derivations and theory.

**Review Assessment: Checking Correctness Of Experiments:**

N/A

**Review Assessment: Thoroughness In Paper Reading:**

N/A

---

> ### Author Response · Authors · 2019-11-10
> **Response to Reviewer 4**
>
> First of all, we thank the reviewer for giving detailed technical comments. The major concern of the reviewer is that there exists simple reduction from infinite-horizon setting to the episodic setting; and it is straightforward to generalize Jin et al (2018) to obtain our results.
>
> However, the reduction given by the reviewer is incorrect. Below we explain why the reduction doesn’t hold, and this clearly illustrates the subtle differences between the infinite-horizon setting and the episodic case. In fact, as we already emphasized in the paper (Section 3.2), infinite-horizon setting cannot be solved by reducing to finite-horizon setting as long as we consider sample complexity instead of regret.
>
> We focus on sample complexity of exploration in infinite horizon setting, which is a standard measure widely used in previous results in this setting, while Jin et al proved a regret bound. Please note that sublinear regret does NOT imply finite sample complexity of exploration.
>
> The reduction giving by the reviewer does NOT work for infinite horizon setting.
> 1.	The algorithm in Jin et. al finds a time-dependent policy. Running finite-horizon algorithm can only guarantee near-optimal behavior at step 1, 1+R, 1+2R, etc. For other steps, the policy given by Jin et. al can be suboptimal. For example, at step 1+R/2, the policy given by finite horizon algorithm only maximizes over the reward of remaining R/2 steps, which cannot guarantee optimal bound in the infinite horizon setting.
> 2.	Sublinear regret does NOT imply finite sample complexity of exploration. For example, if an algorithm takes sub-optimal moves at step 1, 4, 9, …, t^2, …, the regret is bounded by $\sqrt{T}$. However, the sample complexity of exploration of this algorithm is unbounded. Therefore, no direct reduction can be made from sample complexity of exploration to regret.
>
>
> Response to other comments:
>
> Regarding the dependence of  1-gamma:
> Previously best-known model-free algorithm for infinite horizon setting is Delayed Q-learning, which achieves a bound of $\tilde{O}(\frac{SA}{\epsilon^4(1-\gamma)^8})$. We also show that Delayed Q-learning cannot achieve near-optimal bound due to the inefficient usage of samples in Appendix D. Compared to this bound, our result is a significant improvement since we match the lower bound in terms of epsilon as well as S and A up to logarithmic factors. Besides, The previously best claimed result of model-based algorithms is 1/(1-gamma)^6, which is close to our result (1/(1-gamma)^7) and also significantly above the lower bound. Further improving the dependence on 1-gamma is a future direction.
>
> Regarding the technical novelty:
> As stated in section 3.2, there are two major difficulties. Firstly, since we need to bound sample complexity of exploration, we need to establish convenient sufficient condition for being epsilon-optimal. We carefully design Condition 1 and Condition 2 to solve this problem. Secondly, we need to decompose errors to that of non-consecutive steps without any structure. See section 3.2 for detailed discussion.

---

> > ### Comment · AnonReviewer4 · 2019-11-13
> > **Thanks for explanation**
> >
> > Thanks for the explanation. I now agree that the technique sufficiently differs from Jin et al.
> > This is mainly due to the fact that you need to run the algorithm forever and there is no hope to restart even if you stuck at some bad state (this also demonstrates that the def of PAC is not very good ...).
> >
> > >>The algorithm in Jin et. al finds a time-dependent policy. Running finite-horizon algorithm can only guarantee near-optimal behavior at step 1, 1+R, 1+2R, etc. For other steps, the policy given by Jin et. al can be suboptimal. For example, at step 1+R/2, the policy given by finite horizon algorithm only maximizes over the reward of remaining R/2 steps, which cannot guarantee optimal bound in the infinite horizon setting.
> >
> > Actually I do not agree with this. You can set your Q-value in the last stage to be the Q-value of the first stage of the previous iteration. Then even the middle ones (1+R/2) are guaranteed to have near-optimal actions.
> >
> >
> > >>Sublinear regret does NOT imply finite sample complexity of exploration. For example, if an algorithm takes sub-optimal moves at step 1, 4, 9, …, t^2, …, the regret is bounded by . However, the sample complexity of exploration of this algorithm is unbounded. Therefore, no direct reduction can be made from sample complexity of exploration to regret.
> >
> > I would guess you can use the same argument in your paper to bound the number of errors made by the regret algorithm.

---

> > > ### Author Response · Authors · 2019-11-15
> > > **Thanks for the response and new comments**
> > >
> > > We sincerely thank the reviewer for the quick response and new comments.
> > >
> > > Regarding the definition of PAC in infinite horizon settings: Actually this has been extensively discussed in the past two decades. It is believed that sample complexity of exploration is the most natural PAC measurement in this setting. Plese see [1, 2, 3] (below) for more detailed discussion on the performance measurement in infinite horizon settings.
> > >
> > >
> > > >>Actually I do not agree with this. You can set your Q-value in the last stage to be the Q-value of the first stage of the previous iteration. Then even the middle ones (1+R/2) are guaranteed to have near-optimal actions.
> > >
> > > If  we understand correctly, the reviewer is proposing replacing Q-value of the last stage by Q-value of the first stage in the algorithm. (Pleasae correct us if our understanding is wrong.) However, the resulting Q-learning algorithm does not converge. This is because in the finite horizon algorithm there is no discount factor, and replacing the Q-value in the last stage will lead to an unbounded increase in Q-value. If algorithm is modified so that discount factor is included, then the algorithm will be like the one we proposed in the paper. In this case, it is possible to bound the number of sub-optimal steps of this algorithm using our analysis framework. However, this approach is no longer a black box reduction to finite horizon setting because the algorithm is changed.
> > >
> > >
> > > >>I would guess you can use the same argument in your paper to bound the number of errors made by the regret algorithm.
> > >
> > > Yes, we agree that the argument in our paper (namely carefully designed Condition 1 and Condition 2) can be used to bound the number of errors. This argument is one of the technical contribution of our paper.
> > >
> > > [1] Alexander L Strehl and Michael L Littman. An analysis of model-based interval estimation for markov decision processes. Journal of Computer and System Sciences, 74(8):1309–1331, 2008.
> > > [2] Kakade, Sham Machandranath. On the sample complexity of reinforcement learning. Diss. University of London, 2003.
> > > [3] Alexander L Strehl, Lihong Li, Eric Wiewiora, John Langford, and Michael L Littman. Pac model- free reinforcement learning. In Proceedings of the 23rd international conference on Machine learning, pp. 881–888. ACM, 2006.

---

### Decision · Program_Chairs · 2019-12-19

**Decision:**

Accept (Poster)

**Comment:**

In this paper, the authors extended Q-learning with UCB exploration bonus by Jin et al. to infinite-horizon MDP with discounted rewards without accessing a generative model, and proved nearly optimal regret bound for finite-horizon episodic MDP. The authors also proved PAC-type sample complexity of exploration, which matches the lower bound up to logarithmic factors. Overall this is a solid theoretical reinforcement learning work.  After author response, we reached a unanimous agreement to accept this paper.